# Improving a Deep Learning Model to Accurately Diagnose LVNC

**DOI:** 10.3390/jcm12247633

**Published:** 2023-12-12

**Authors:** Jaime Rafael Barón, Gregorio Bernabé, Pilar González-Férez, José Manuel García, Guillem Casas, Josefa González-Carrillo

**Affiliations:** 1Computer Engineering Department, University of Murcia, 30100 Murcia, Spain; jrafael.baron@um.es (J.R.B.); pilargf@um.es (P.G.-F.); jmgarcia@um.es (J.M.G.); 2Hospital Universitari Vall d’Hbron, 08035 Barcelona, Spain; gcasasmasnou@gmail.com; 3Hospital Virgen de la Arrixaca, 30120 Murcia, Spain; josegonca.alarcon@gmail.com

**Keywords:** left ventricular non-compaction diagnosis, cardiomyopathies, convolutional neural networks, MRI Image segmentation

## Abstract

Accurate diagnosis of Left Ventricular Noncompaction Cardiomyopathy (LVNC) is critical for proper patient treatment but remains challenging. This work improves LVNC detection by improving left ventricle segmentation in cardiac MR images. Trabeculated left ventricle indicates LVNC, but automatic segmentation is difficult. We present techniques to improve segmentation and evaluate their impact on LVNC diagnosis. Three main methods are introduced: (1) using full 800 × 800 MR images rather than 512 × 512; (2) a clustering algorithm to eliminate neural network hallucinations; (3) advanced network architectures including Attention U-Net, MSA-UNet, and U-Net++.Experiments utilize cardiac MR datasets from three different hospitals. U-Net++ achieves the best segmentation performance using 800 × 800 images, and it improves the mean segmentation Dice score by 0.02 over the baseline U-Net, the clustering algorithm improves the mean Dice score by 0.06 on the images it affected, and the U-Net++ provides an additional 0.02 mean Dice score over the baseline U-Net. For LVNC diagnosis, U-Net++ achieves 0.896 accuracy, 0.907 precision, and 0.912 F1-score outperforming the baseline U-Net. Proposed techniques enhance LVNC detection, but differences between hospitals reveal problems in improving generalization. This work provides validated methods for precise LVNC diagnosis.

## 1. Introduction

Nowadays, one of the leading causes of death globally is cardiovascular diseases that cause about 32% of all deaths worldwide [1,2]. Therefore, early detection of cardiac anomalies is paramount for a favorable patient treatment outcome. Among these diseases, Left Ventricular Non-Compaction Cardiomyopathy (LVNC) is a rare cardiac condition characterized by an abnormally spongy and thickened left ventricular wall, in contrast to the typical smooth and firm structure. The significance of LVNC lies in its potential to induce a broad spectrum of symptoms, ranging from fatigue and shortness of breath to the development of heart failure. Moreover, it is associated with an increased risk of suffering from other diseases such as HCM (Hypertrophic Cardiomyopathy), DCM (Dilated Cardiomyopathy), and ARVD (Arrhythmogenic Right Ventricular Cardiomyopathy).

Currently, there is no consensus on what is considered LVNC [3,4,5,6]. For this reason, we commonly use Petersen’s criteria [7]. One distinctive feature of LVNC is the abundance of trabeculae within the left ventricle. We can use the percentage of trabecular volume (VT%) to diagnose whether or not a patient has LVNC. To do this, we choose a threshold of VT% over which you are considered to have LVNC. In this light, QLVTHC is a fully automatic tool that uses a threshold of 27.4% for diagnosis [8,9,10].

Deep Neural Networks (DNNs) have gained immense popularity in recent years and are now considered one of the primary methods of modern Artificial Intelligence (AI) [11]. DNNs are being utilized across various scientific domains such as image recognition, speech recognition, autonomous vehicles, and medical research [12,13,14,15]. New automatic methods based on DNNs have been developed to determine the left ventricle volume through Magnetic Resonance Imaging (MRI) [16,17]. Recent publications also exploit various possibilities of deep learning to segment the left and the right ventricle by using high-performance computing [18,19,20,21,22]. In particular, DL-LVTQ is a new automatic proposal based on a U-Net architecture [23] to diagnose LVNC. These new tools offer several advantages for medical professionals and healthcare institutions:Efficiency: automated systems can quickly analyze a large dataset, allowing clinicians to focus on complex cases and treatment plans.Standardization: the use of machine learning algorithms ensures a uniform criterion for diagnosis, mitigating the risk of human error and subjective bias.Accessibility: such automated tools make expert-level diagnostics more readily available, particularly in settings with limited access to specialized medical professionals.Continual Learning: These systems can adapt and improve their diagnostic capabilities over time by learning from new data, which are especially beneficial for identifying atypical or rare cases.Resource Allocation: Automation can free up medical resources, allowing healthcare providers to allocate their focus and finances more effectively.

It is important to point out that LVNC is a rare disease (8–12 individuals per million inhabitants), and having the ability to diagnose the condition automatically upon leaving the MRI machine is invaluable. This not only enables the detection of incidental cases that were not the original focus of the MRI but also expedites the identification of cases that were intentionally being investigated. As a result, it streamlines the allocation of healthcare professionals’ time, frequently wasted on excluding this rare condition from the list of potential diagnoses.

Various methodologies, such as cardiac volume quantification using U-Net [24], along with “Active Shape Models” in ultrasound imaging [25], illustrate the breadth of current techniques. Further amplifying this range, Ref. [26] explores the application of advanced segmentation methods for 3D heart reconstruction.

This paper aims to improve upon the results obtained in [27] using the same database. To achieve this, we perform the following proposals:Utilize the original 800 × 800 size of the images as input for the neural net instead of the reduced 512 × 512 size used in [27].Employ a clustering algorithm to eliminate hallucinations from the U-Net output. This clustering algorithm improves the robustness of the results by using all the segmentations produced by the U-Net of a patient to ensure that the selected area corresponds to the left ventricle.Try out three different neural net architectures: AttU-Net, MSA-UNet, and U-Net++ [28,29,30]. All of them are variations of the U-Net.To make the most of transfer learning, we will fine-tune each neural network to fit each subset of the data. Since the data used for training come from distinct sources, this approach allows us to optimize our performance for each neural net and subset of data.

## 2. Materials and Methods

We aim to diagnose whether a patient has LVNC by using convulational neuronal networks (CNNs). We diagnose LVNC by obtaining the percentage of trabecular volume:(1)$$\mathrm{VT}\%=\frac{\mathrm{Trabecular}\;\mathrm{volume}}{\mathrm{Trabecular}\;\mathrm{volume}+\mathrm{Compacted}\;\mathrm{volume}}\cdot 100.$$

The output of our neural net looks like Figure 1, where we color three different zones: External Layer (EL), Internal Cavity (IC), and Trabecular Zone (TZ) in three different colors. Our U-Net-based proposal takes this output and calculates the VT%, giving us the access to determine if the patient has LVNC.

Three hospitals were involved in the study—Virgen de la Arrixaca of Murcia (HVAM), Mesa del Castillo of Murcia (HMCM), and Universitari Vall d’Hebron of Barcelona (HUVHB). HVAM has two scanners from Philips and General Electric, both of 1.5 T. The acquisition matrices of the scanners are 256×256 pixels and 224×224 pixels with a pixel spacing of 1.5×1.5×0.8 mm and 1.75×1.75×0.8 mm, respectively. The General Electric model scanner used by HMCM is the same as that used by HVAM. HUVHB uses a 1.5 scanner Avanto of Siemens, where the acquisition matrix is 224×224 pixels.

The LV function is determined with balanced steady-state free precision (b-SSFP) sequences, where the repetition interval is established to 3.8 ms for HMCM and HUVHB, whereas HVAM uses 3.3 ms. Other parameters like echo time, flip angle, echo train length, slice thickness, slice gap, and phases are fixed to 1.7 ms, $60^{\circ }$, 23, 8 mm, 2 mm, and 20 phases, respectively, for all scanners. All patients were monitored in apnea, in synchronization with the ECG and without a contrast agent.

### 2.1. Populations

The dataset is composed of a subset obtained from HVAM, marked as P, another subset from HMCM, marked as X, and another subset from HUVHB, marked as H. This dataset matches the one used in [27].

A sample image of each dataset is showed in Figure 2. The differing results shown in Table 1 are due to the fact that the P dataset has more images than X and H combined, but also that these datasets are intrinsically different as:The set P consists of 293 patients with hypertrophic cardiomyopathy (HCM), which involves thickening of the heart muscle.The set X is made up of 58 patients with various types of heart diseases, among which HCM is also included.The set H consists of 28 patients diagnosed with LVNC using the Petersen criteria.

To use the same criteria as [27], we use exactly the same 80/20 train/test split and perform five-fold cross-validation on the training dataset. Table 1 shows the Dice coefficients of each test dataset trained on P + X + H. We report the mean and standard deviation of the test data evaluations from the five folds. Because of the large imbalance in the number of images per hospital, we present results separately for each of the three hospitals. Otherwise, if we aggregate the results, those from hospital P would dominate, overshadowing those from hospitals X and H.

Moreover, it is important to note that the dataset is fully valid from a medical point of view, as the output images of the neural network have been positively assessed by several cardiologists in previous works [23,27].

### 2.2. 800 × 800 Images

In the cited U-Net study [27], images were initially upscaled to 800 × 800 at which they were segmented. Afterward, they were resized to 512 × 512 for memory efficiency. In order to leverage the original 800 × 800 segmentations, we propose an initial layer to downsample images to 200 × 200, leveraging the U-Net’s existing “blocks” composed of two 3 × 3 convolutions with batch normalization and leaky ReLU activations (0.01 negative slope). The convolutions have a stride of two, halving each dimension.

The traditional U-Net works on this 200 × 200 input and produces a 200 × 200 output. To obtain an 800 × 800 output, we append an additional decoding layer before the final convolution. This layer consists of two transposed convolutions, each with a stride of two, interspersed with two UNet-like blocks, effectively resizing the image back to 800 × 800.

The complete architecture is depicted in Figure 3.

### 2.3. Clustering Algorithm

After analyzing the results obtained from traditional U-Net, we noticed it exhibited hallucinations. Sometimes, it produces multiple left ventricles, the most common case being mislabeling the right ventricle as the left ventricle. This issue occurs very occasionally with the best networks we have obtained. Trying to solve this issue we propose a two-stage algorithm that improves the results and makes them more robust. We next present this two-stage algorithm.

Our first algorithm focuses on cluster generation within each image, which should represent the left ventricle (Algorithm 1). We create a custom method inspired by K-means. The steps are:

**Algorithm 1** Cluster Generation
Find the three largest clusters in each image, considering any set of pixels marked as EL, IC, or TZ connected as a cluster.Examine clusters for separable IC + TZ  components. If found, reassign EL pixels to the closest new cluster.If clusters fragment, merge any sub-cluster smaller than 5% of the largest cluster with its neighboring clusters, we will consider that the neural network may have incorrectly enclosed the IC + TZ mini-cluster.


With these clusters established, Algorithm 2 selects the most likely left ventricle cluster. The algorithm proceeds in two passes and uses a list of ’skip’ images for ambiguous cases.

**Algorithm 2** Cluster Selection
Find a sequence of clusters (from consecutive images) that minimizes the distance between consecutive centroids.Compute the overlap percentage for each adjacent pair in the sequence.If any pair has an overlap of less than 20%, proceed to step 4; otherwise, go to step 6.Calculate overlap percentages for all cluster pairs (regardless of order). Thus, for each cluster, count how many clusters overlap with it more than 20%.Add to the skip list the first image with the least valid overlaps, then return to step 1.If step 6 was reached before, or there are no images to skip, proceed. Otherwise, mark surviving images as fixed, evidence the skip list, and return to step 1.Output associated cluster for each image; if marked to skip, output blank image (Background), as we will consider that the left ventricle has not been detected.


Figure 4 presents an example of the algorithm in action. Figure 4a shows the original output generated by the neural network, and Figure 4b shows this output overlaid with errors marked in red. As we can see, there are several areas the network identifies as the left ventricle that are not. Each region is marked as an independent cluster by the algorithm. Furthermore, there is an area marked in red, but not entirely because this is the accurate left ventricle. This part of the output is also celebrated as a cluster, and our goal with the algorithm is to choose this. By applying these algorithms, we obtain Figure 4c, which, as we can see in Figure 4d, is the desired cluster and has eliminated problematic clusters.

### 2.4. New Neural Net Architectures

Next, we briefly overview the the new neural networks used in this dataset.

#### 2.4.1. Attention U-Net

The AttU-Net [28] is a U-Net with an attention mechanism shown in Figure 5. Instead of directly concatenating encoder and decoder outputs like in traditional U-Net, we use the attention mechanism’s output to concatenate the encoder’s output.

We obtained the attention mechanism from [31], changing the batch normalization to instance normalization to be like our U-Net. We have also omitted the resampler as in [31].

#### 2.4.2. MSA-UNet

This U-Net aims to be an alternative to the traditional U-Net using the same structure but different operations. The architecture of the U-Net used in the article [29] is represented in Figure 6. We also use this architecture but change batch normalization with instance normalization and bilinear interpolations with transposed convolutions. These changes are made in the various operations performed in this network: MSRB, RASM, AASPP, and MSAM.

#### 2.4.3. U-Net++

U-Net++ is a variation in the traditional U-Net that allows us to use a new training approach. The architecture of this neural network is obtained from the article [30], as described in the diagram in Figure 7.

Apart from all the added connections, the main difference between U-Net++ and the traditional U-Net is the different training strategy known as deep supervision. In this approach, the network is trained to assume that $x^{0,j}\;\forall j>0$ are outputs of the network after the necessary post-processing. All these outputs are concatenated by applying an additional 1 × 1 convolution, which, after post-processing, is the final output of the U-Net++, also used for training.

Since we ultimately want a single output for our U-Net++, but have trained multiple outputs, it is natural to assign weights to the relevance of each of these outputs so that the final output is trained more strongly. Considering that our U-Net++ is like the one in Figure 7, which has $x^{0,j}j\in 1,2,3,4$, we use the following loss function:(2)$$L=0.25L_{1}+0.25L_{2}+0.5L_{3}+0.75L_{4}+L_{output}.$$

Our primary interest is obtaining the best possible output. After this training, we perform another round of training to refine the network, using $L_{output}$ as the loss function.

### 2.5. Transfer Learning

To improve the results in X and H, we fine tune the neural nets trained on the entire dataset (P + X + H) on X and H separately. In doing so, we transfer the general knowledge from across all hospitals to try to improve the results on both X and H. Notice that this means that every neural net will branch into two fine-tuned neural nets.

To fine tune the Unet network, we will freeze the Encoder weights of each Unet. The Encoder compresses the input image size while expanding the number of channels, creating a condensed representation of the original data. It is important to note that even though the U-Net++ network is more complex, it still has an Encoder that is formed by the path composed of $X^{i,0}\;\forall i$.

## 3. Results and Discussion

This section presents the influence of increasing the image size from 512 × 512 to 800 × 800 and of the proposed clustering algorithm on the results obtained by a U-Net. Then, we provide the results achieved by the other neural net architectures (AttU-Net, MAS-UNet, and U-Net++) and compare them with those obtained by a regular U-Net.

### 3.1. 800 × 800 Images and Clustering Algorithm

Using 800 × 800 images and the clustering algorithm increased the Dice coefficient, whose values we show in Table 2.

As we can see, the net result is an overall increase of around 0.02 in the Dice coefficient. The clustering algorithm has a much lesser impact than the average Dice coefficient as it only affects a small number of images, and most hallucinations are slight. This is clearer in Figure 8, where we show only the changes produced in the U-Net. Additionally, we see that some cases with very low Dice coefficients achieve a Dice coefficient of 0. This means the cluster was not found, possibly because the results were inferior.

On the other hand, it is important to emphasize that the vast majority of improvements (green dots) are substantial, averaging around 0.06 in the Dice coefficient for the trabecular zone. This is a significant improvement, as the benefit we gained from using 800 × 800-sized images is only around 0.02 in the Dice coefficient. Figure 8 shows that most changes are found in the external layer. This is likely because there must be an outer layer to have trabeculae or an internal cavity, so it is easier to have just EL than to have all three components. Furthermore, it is relevant to point out that the hallucination issue decreases dramatically when using the other networks, dropping from 146.6 images per fold (each fold has 602 images) to only about 17 images per fold.

### 3.2. Model Comparison

To compare the results of the different neural networks, we display in Table 3, Table 4 and Table 5 the average Dice coefficients of the five folds on the test set.

In the following tables, we refer to U-Net++ as U-Net++* after it has been trained with the weights shown in (Equation 2). Thus, after refining it by training only with the $L_{output}$ error function, we refer to it as U-Net++.

From Table 3, Table 4 and Table 5, it is clear that U-Net++ outperforms U-Net++* in all cases, indicating that the refinement was beneficial. Furthermore, U-Net++ outperforms the other U-Nets in datasets P and X across all categories. Interestingly, the H dataset loses in all categories against the other U-Nets. This may indicate that its high complexity prevents it from generalizing well to other datasets, as dataset H has the fewest images, using only 16 patients for training.

Regarding the attention mechanism-based networks (AttUNet and MSA-UNet), we see that they behave very differently. On the one hand, the AttUNet gives results similar to the traditional U-Net, with significant differences only in dataset H, where the traditional U-Net performs better. This means this implementation is useless as a simpler version outperforms it.

On the other hand, the MSA-UNet appears to be a strong competitor to U-Net++, achieving similar results in all three datasets. However, it significantly outperforms in dataset H. This suggests that it is better to start with the MSA-UNet when generalizing to a new hospital. Still, once a sufficiently large database is established, the U-Net++ gains enough traction to learn correctly.

### 3.3. Transfer Learning

This section aims to improve the U-Net’s results in datasets X and H, possibly equivalent to P’s. Since the AttUNet performed poorly earlier, we excluded it from this analysis and focused on the traditional U-Net, MSA-UNet, and U-Net++. By freezing the encoder parameters and fine tuning on X (training on X and evaluating on X), we obtain Table 6. Similarly, we obtain the results for H in Table 7.

As we can see, the improvements brought about by transfer learning in X are almost negligible. This can be seen in comparing Table 4 and Table 6, with the maximum difference between the Dice coefficients being 0.004. On the other hand, there are significant changes in dataset H, with the MSA-UNet’s Dice coefficient in the trabecular zone improving by 0.013.

For this reason, we do not use the networks obtained through transfer learning to differentiate the datasets in U-Net++ or datasets P and X in the MSA-UNet. However, we determine dataset H for the MSA-UNet.

### 3.4. LVNC Detection

For LVNC detection, we use Equation (Equation 1) with a threshold of 27.4%; i.e., any patient with a VT% higher than 27.4% is classified as having LVNC. In this section, we examine the predictive ability of U-Net++ for LVNC, as it performed best in datasets P and X. Following this analysis, we examine the results on dataset H for both U-Net++ and MSA-UNet to determine if there is a significant improvement.

#### U-Net++

A confusion matrix allows us to statistically evaluate the network model from a medical point of view. Next, we show the results of the U-Net++ while also using Algorithm 2 in Figure 9b. To compare with the results of the article [27], we display their results in Figure 9a.

We can observe that the new model detects fewer patients with LVNC, the positive predictive value decreases slightly from 0.94(210/223) to 0.91(204/223), than the traditional U-Net, but the negative predictive value has a substantial increase from 0.78(122/156) to 0.87(135/156), enlarging by 13 the detection of healthy people. However, it achieves higher accuracy, precision, and F1-score (Table 8), resulting in a much better-prepared model for several types of cardiomyopathies. Therefore, a cardiologist can receive an automatic diagnosis almost instantly and with excellent reliability. Nonetheless, it has a slightly lower recall, which is an essential measure in our case, as it indicates the cases with LVNC that are not detected.

### 3.5. Areas of Improvement

To identify future research options for improving LVNC detection models, we should determine where our model is failing.

#### 3.5.1. Dice-Area Relationship

We analyzed images with low Dice coefficients and discovered they are often found at the extremities of a patient’s magnetic resonance slices. We used normalized slice numbers for this analysis:$$\mathrm{Normalized}\;\mathrm{Slice}=\frac{\mathrm{Slice}-\min({\mathrm{Slices}}_{\mathrm{patient}})}{\max\left({\mathrm{Slices}}_{\mathrm{patient}}\right)-\min\left({\mathrm{Slices}}_{\mathrm{patient}}\right)}$$

Thus, a slice is at an end of its value is 0 or 1. 

We observed that slices with low Dice coefficients (<0.5) commonly occur at the ends and usually display limited trabeculation (Figure 10). Three main factors could contribute to this:Due to few trabeculae, small errors drastically decrease the Dice coefficient.Edge effects have a much greater significance; the neural network often creates a mini-layer of trabeculae between the inner and outer layers that humans probably would not mark.The database primarily contains images of hearts with hypertrophic cardiomyopathy (set P), which are enlarged. Due to the lack of available small heart images, accurate segmentation is difficult.

Interestingly, the Dice coefficient’s variance is not as pronounced for the outer layer and inner cavity. This may be due to the trabecular zone’s circular crown shape, making it sensitive to minor changes, whereas Figure 11 shows a correlation between low Dice coefficients and low areas, particularly for the trabecular zone, the generally smaller areas in this zone prevent definitive conclusions about the impact of circular crown effects.

#### 3.5.2. Differences between Hospitals

Before starting this work, one of our expectations was to find a neural network that would generalize better to other sets (improve in X and H) or that by using Transfer Learning from the already trained networks, we would achieve similar results in all hospitals. However, although the methods used have improved results in all sets, even bringing set X to levels higher than the initial ones in P, there is still a noticeable difference in the results from different hospitals. This means that by increasing the number of images from X and H, we should be able to improve results in those sets, reaching P levels.

## 4. Conclusions

This paper improves a deep learning approach to left ventricular trabecular quantification based on a U-Net architecture [27] by leveraging the original 800 × 800 images and a clustering algorithm. This modification leads to an improvement of around 0.02 average trabecular zone Dice coefficient. Furthermore, we also analyze the behavior of more advanced neural net architectures such as AttU-Net, MAS-UNet, and U-Net++. These neural net architectures contribute a cumulative 0.04 boost in the average trabecular zone Dice coefficient.

Even though deep learning approaches to automatically diagnose LNVC cardiomyopathy are complex, we consider our results promising. Indeed, our proposal allows cardiologists to infer patients without significant effort and personal subjectivity. Although there is still room for improvement, an automated, fast, and robust system for determining LVNC will provide cardiologists with a diagnosis without spending considerable time, eliminating human error and subjectivity. Proposals like ours favor the evolution of LVNC and other heart diseases. However, it is interesting to note that among all parties, this approach will primarily benefit the patient.

The dataset is imbalanced, favoring images of left ventricles with extensive trabeculae. This imbalance likely contributes to the model’s poorer performance on images featuring low trabeculae. In future studies, we plan to expand our dataset, obtaining new images to solve this imbalance.

## Figures and Tables

**Figure 1 jcm-12-07633-f001:**
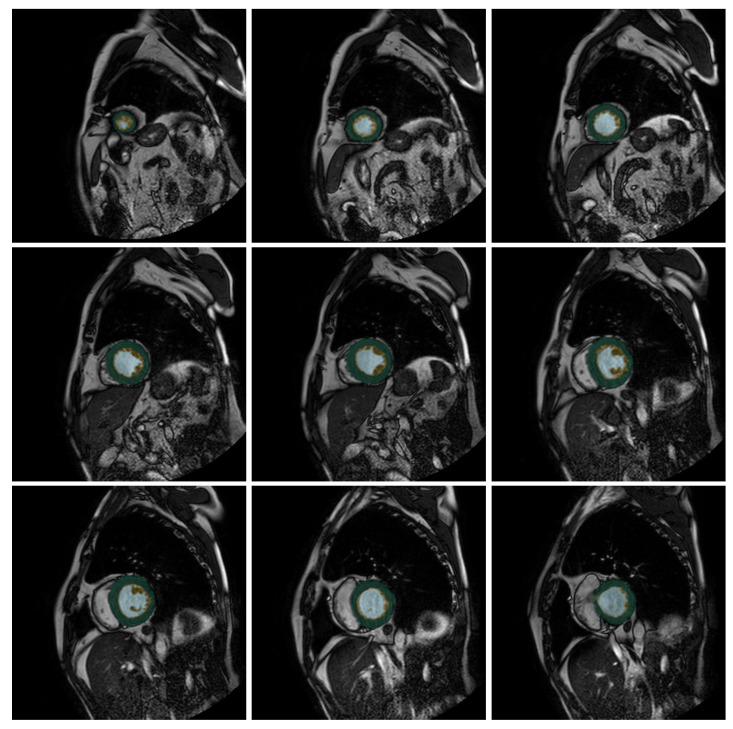
Output slices for the patient P241 (from the test set) from the U-Net++. Green indicates the compacted external layer of the left ventricle, yellow the trabecular zone, and light blue the internal cavity.

**Figure 2 jcm-12-07633-f002:**
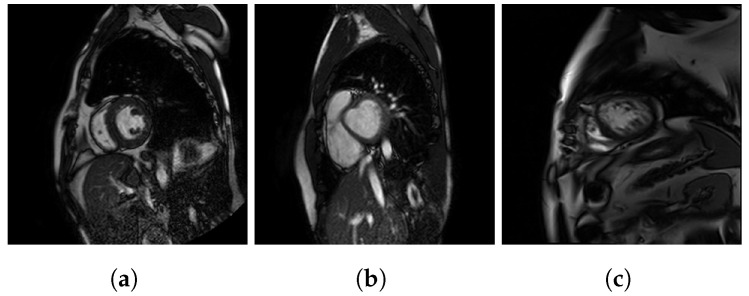
Sample images of each dataset: (**a**) P dataset, (**b**) X dataset, and (**c**) H dataset.

**Figure 3 jcm-12-07633-f003:**
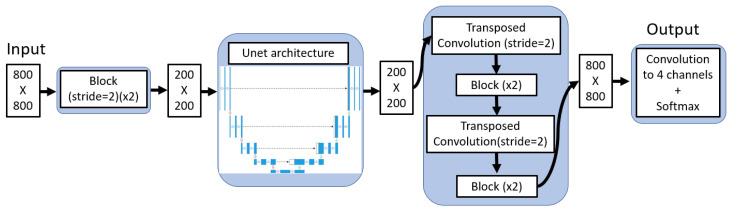
Diagram of the entire 800 × 800 image neural network.

**Figure 4 jcm-12-07633-f004:**
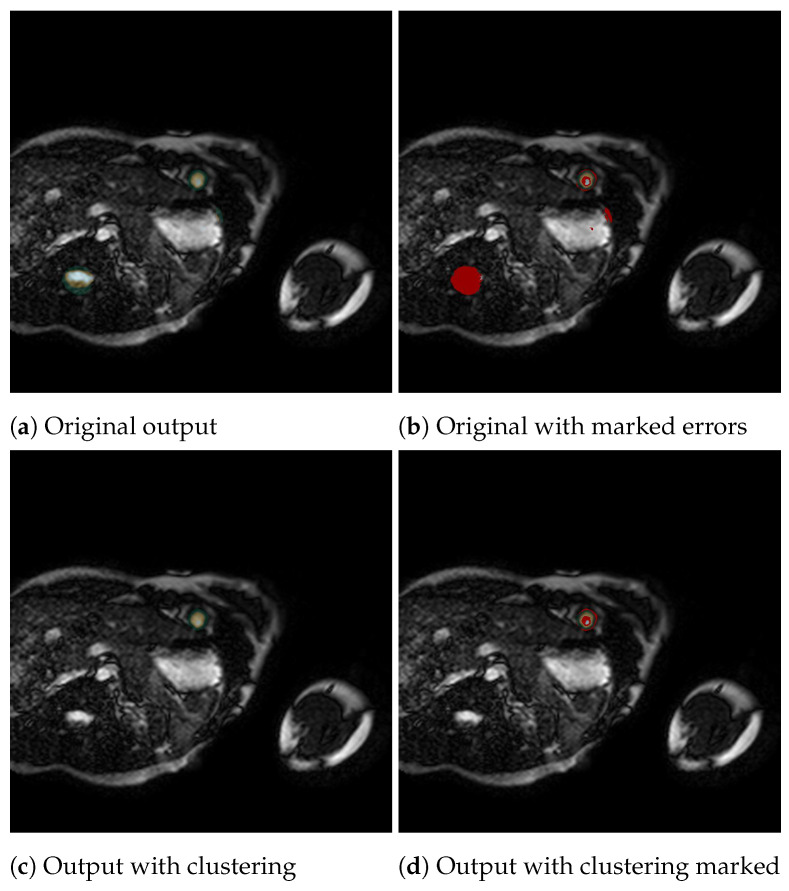
Example of the clustering algorithm improving an image, showing in red in (**b**,**d**) the errors of the corresponding outputs (**a**,**c**).

**Figure 5 jcm-12-07633-f005:**
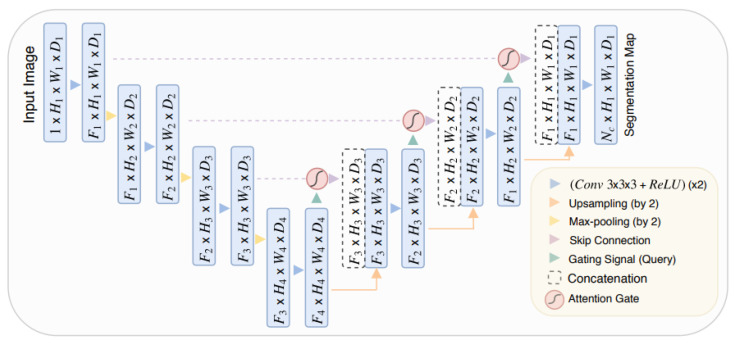
Attention U-Net used in the article [28].

**Figure 6 jcm-12-07633-f006:**
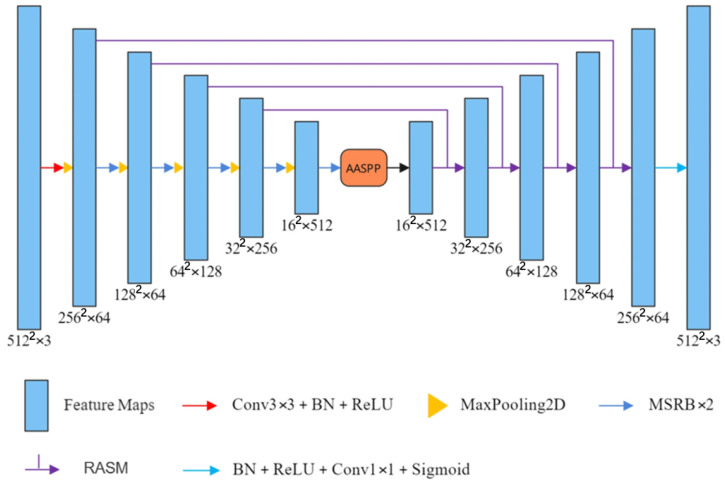
MSA-UNet modified from [29].

**Figure 7 jcm-12-07633-f007:**
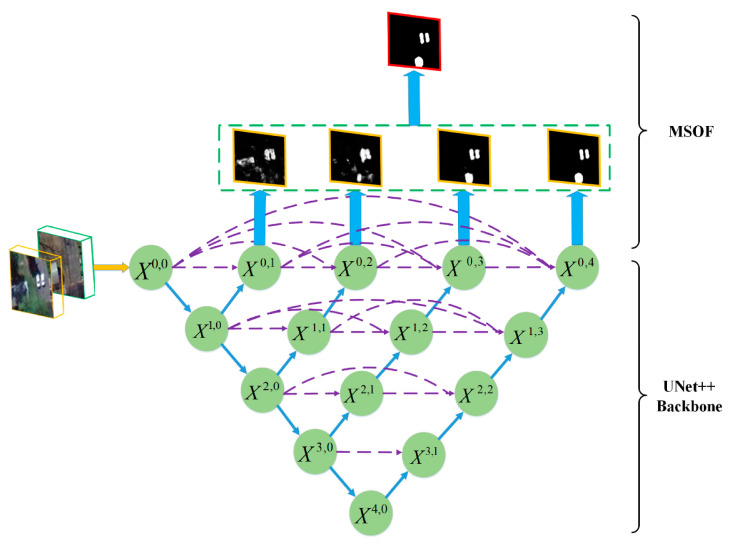
Basic architecture of U-Net++ with an additional layer for output fusion called MOST (Multiple Side-Output Fusion) for training modified from [30].

**Figure 8 jcm-12-07633-f008:**
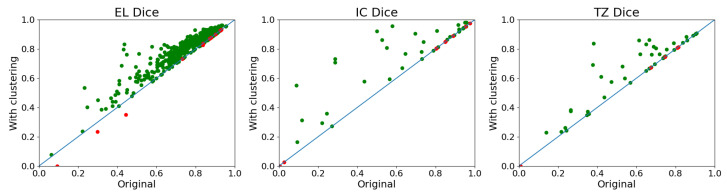
This graph displays the changes that occurred in the U-Net due to the effects of Algorithm 2 in the five folds simultaneously. The same image can appear multiple times. The unaltered values are shown on the “x” axis, and the results after clustering are shown on the “y” axis. Improved results are indicated in green, whereas worsened results are shown in red. The blue line is meant to help distinguish improvements from worsening.

**Figure 9 jcm-12-07633-f009:**
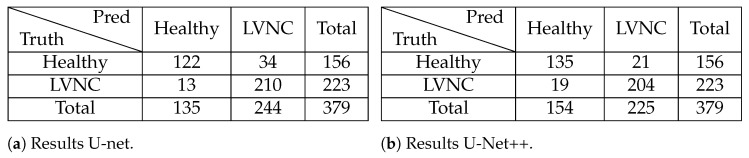
(**a**) Confusion matrix for LVNC detection from the article [27]. (**b**) Mean of the confusion matrices obtained from 5 to fold cross-validation using U-Net++ on the entire dataset (P + X + H) with a threshold of 27.4%. The standard deviation across all values is approximately 3 or 4 patients, omitted for simplicity in comparison.

**Figure 10 jcm-12-07633-f010:**
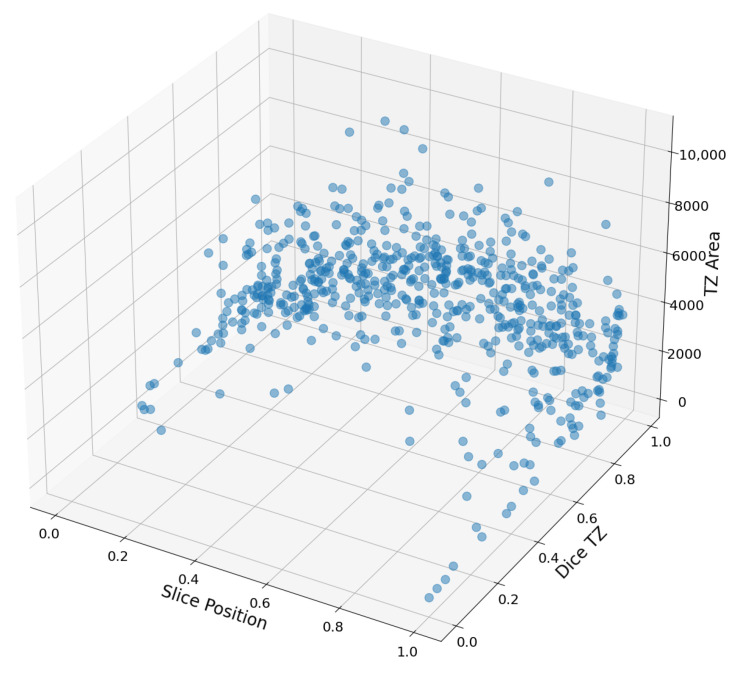
Determine the Dice coefficient between the area of a given section (measured in pixels) and its corresponding image number.

**Figure 11 jcm-12-07633-f011:**
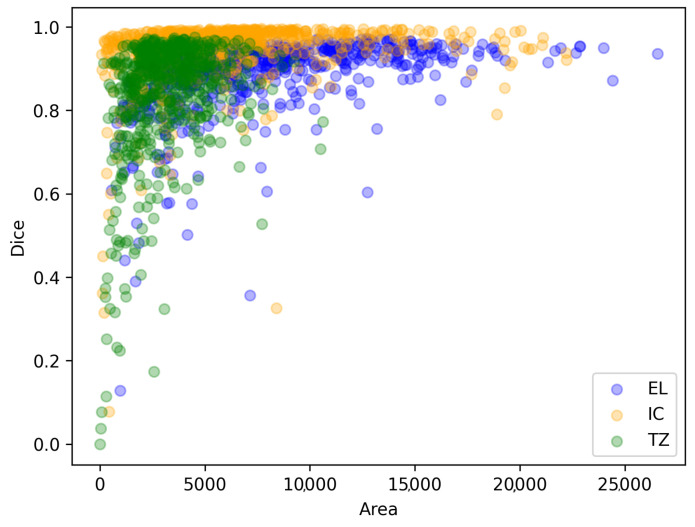
Dice coefficient against the area of that section (measured in number of pixels).

**Table 1 jcm-12-07633-t001:** Average (±standard deviation across the five folds) of the Dice coefficient in [27] of each test dataset trained on P + X + H, where EL is the External Layer, IC is the Internal Cavity, and TZ is the Trabecular Zone.

Population	Dice EL	Dice IC	Dice TZ	Average Dice
P	0.89±0.09	0.94±0.09	0.84±0.14	0.89±0.09
X	0.84±0.14	0.93±0.14	0.80±0.18	0.86±0.13
H	0.86±0.09	0.92±0.10	0.79±0.16	0.86±0.09

**Table 2 jcm-12-07633-t002:** Average (±standard deviation across the five folds) of five folds on the test set after applying the clustering algorithm and after training on images of P + X + H, where EL is the External Layer, IC is the Internal Cavity, and TZ is the Trabecular Zone.

Population	Dice EL	Dice IC	Dice TZ	Average Dice
P	0.893±0.008	0.954±0.003	0.852±0.008	0.900±0.005
X	0.861±0.005	0.9622±0.0018	0.830±0.007	0.884±0.004
H	0.864±0.006	0.940±0.006	0.813±0.010	0.873±0.006

**Table 3 jcm-12-07633-t003:** Evaluation on P.

Red Neuronal	Dice EL	Dice IC	Dice TZ	Average Dice
U-Net	0.893±0.008	0.954±0.003	0.852±0.008	0.900±0.005
AttUNet	0.888±0.009	0.9542±0.0023	0.847±0.011	0.896±0.007
MSA-UNet	0.906±0.004	0.957±0.003	0.866±0.005	0.910±0.004
U-Net++*	0.908 ± 0.003	0.9574 ± 0.0015	0.8670 ± 0.0023	0.9107 ± 0.0020
U-Net++	0.9122±0.0011	0.9592±0.0008	0.8704±0.0011	0.9139±0.0006

**Table 4 jcm-12-07633-t004:** Evaluation on X.

Red Neuronal	Dice EL	Dice IC	Dice TZ	Average Dice
U-Net	0.861±0.005	0.9622±0.0018	0.830±0.007	0.884±0.004
AttUNet	0.854±0.006	0.963±0.003	0.825±0.011	0.881±0.005
MSA-Unet	0.874±0.006	0.9648±0.0016	0.850±0.005	0.896±0.003
U-Net++*	0.874 ± 0.007	0.9658 ± 0.0011	0.848 ± 0.005	0.896 ± 0.004
U-Net++	0.884±0.003	0.9672±0.0016	0.855±0.003	0.9019±0.0020

**Table 5 jcm-12-07633-t005:** Evaluation on H.

Red Neuronal	Dice EL	Dice IC	Dice TZ	Average Dice
U-Net	0.864±0.006	0.940±0.006	0.813±0.010	0.873±0.006
AttUNet	0.847±0.005	0.935±0.004	0.794±0.008	0.859±0.005
MSA-Unet	0.878±0.005	0.938±0.005	0.816±0.012	0.877±0.006
U-Net++*	0.869 ± 0.011	0.923 ± 0.011	0.804 ± 0.012	0.8652 ± 0.010
U-Net++	0.877±0.008	0.928±0.008	0.809±0.012	0.871±0.009

**Table 6 jcm-12-07633-t006:** Transfer learning in X after applying Algorithm 2.

Red Neuronal	Dice EL	Dice IC	Dice TZ	Dice Promedio
U-Net	0.865±0.005	0.963±0.0024	0.8346±0.008	0.888±0.005
MSA-UNet	0.877±0.006	0.9652±0.0016	0.849±0.006	0.897±0.004
U-Net++	0.883±0.003	0.9668±0.0008	0.853±0.003	0.9009±0.0019

**Table 7 jcm-12-07633-t007:** Transfer learning in H after applying Algorithm 2.

Red Neuronal	Dice EL	Dice IC	Dice TZ	Dice Promedio
U-Net	0.866±0.007	0.943±0.006	0.819±0.011	0.876±0.007
MSA-UNet	0.882±0.004	0.946±0.004	0.828±0.009	0.885±0.005
U-Net++	0.878±0.008	0.929±0.007	0.812±0.011	0.873±0.008

**Table 8 jcm-12-07633-t008:** Comparison of metrics with [27] for our new U-Net.

	U-Net [27]	U-Net++
Accuracy	0.876	0.896
Recall	0.942	0.917
Precision	0.861	0.907
F1-score	0.899	0.912

## Data Availability

The dataset used to support the findings of this study was approved by the local ethics committee and so cannot be made freely available. Requests for access to these data should be made to the corresponding author, Gregorio Bernabé, gbernabe@um.es.

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
