# Peer review of "Improving a Deep Learning Model to Accurately Diagnose LVNC"

_jcm, 2023, doi:10.3390/jcm12247633_

Round 1
Reviewer 1 Report
Comments and Suggestions for Authors
This paper aims to improve the deep learning model for the accurate diagnosis of LVNC. This study is sound. However, there are some concerns about this study.
- The main technical challenge of the proposed method should be clarified
- The claimed contributions require more explanations. For example, why it is needed to consider 800*800 size and 500*500 size of the images. Why consider the variants of U-net ?
- "dataset a mix of images obtained from three hospitals and 85
different scanners". Please provide the information of different scaners.
- More studies on cardiac image analysis need to be cited to enhance the literature review, e.g.: Region-of-interest-based cardiac image segmentation with deep learning; Causal knowledge fusion for 3d cross-modality cardiac image segmentation; Deep neural network architectures for cardiac image segmentation.
- What is the main modification of U-net proposed in this study ?
- Figure 1 occupies too much space.
- Section 2.4 is too brief. More details should be provided.
- In Figure 7, why are the most points in the right uppermost corner ?
- Figure 9 is difficult to observer clearly. Please refine this figure.
- Some grammatical errors and typos.
Comments on the Quality of English Languagemoderate refinement
Reviewer 2 Report
Comments and Suggestions for Authors
I would like to inform the authors that this manuscript holds intrinsic value and possesses high scientific significance. While the methodology is acceptable, some improvements in the organization and content are necessary. The following suggestions can enhance the scientific value of your manuscript:
1. Abstract: Revise the abstract to present the introduction, method, results, and conclusion in separate paragraphs. This will improve the readability and structure of your manuscript. Additionally, highlight the main findings of your study, particularly the metrics of the proposed model.
2. Background: It seems that you transitioned to the main subject without providing sufficient background information. This reduces the readability of your manuscript. I recommend including references that discuss the application of artificial intelligence in LVNC diagnosis to help readers establish a better understanding of your methodology. Consider incorporating the following references to enrich your manuscript's content:
- "Efficient framework for detection of COVID-19 Omicron and delta variants based on two intelligent phases of CNN models"
- "A mobile application based on an efficient lightweight CNN model for the classification of B-ALL cancer from non-cancerous cells: a design and implementation study"
- "Clinical decision support system for early detection of prostate cancer from benign hyperplasia of the prostate"
- "A hybrid particle swarm and neural network approach for the detection of prostate cancer from benign hyperplasia of the prostate"
Including these references will provide suitable sources for your manuscript's readers and enhance the richness of your reference list.
3. Dataset Section: Separate the dataset section and include specifications and images of the dataset in that section.
4. Section 2.4: In this section, you mentioned "transfer learning," but only discussed parameter tuning. It is recommended to logically explain the concept of transfer learning and mention it in the introduction section. If necessary, elaborate on transfer learning in Section 3.3. Alternatively, you may choose to remove this reference to maintain clarity.
5. Section 3.4.1: Ensure that the image captions are correctly provided. Include the appropriate legend for the image in Section A and adjust its font to differentiate it from the main text. It is preferable to present the confusion matrix in its original form as an image rather than as a table (optional).
6.Share the implementation code of your model project so that we can check it.
By incorporating these suggestions, you can enhance the overall quality and readability of your manuscript while adding to its scientific value.
Reviewer 3 Report
Comments and Suggestions for Authors
An interesting, educational and informative manuscript that has clinical merit. However, there are editing issues that the authors should consider and address. The following are suggestions/comments regarding those issues. The authors should evaluate the verb tense utilized in areas throughout the manuscript. Lines 3 & 4, "Trabeculated left ventricle indicates LVNC, but ...". Line 13, "... but differences occur across hospitals ...". Line 26, "of breath to the development of heart failure." Line 81, "... where we color three different zones: External Layer (EL), Internal Cavity (IC) and Trabecular Zone (TZ) in three ...". Lines 83 & 84, "... giving us the access to determine if the patient ...". Line 136, "... new neural networks used in this ...". Line 139, "... U-Net with an attention mechanism ...". Line 160, "... 1x1 convolution; which, after ...". Line 163, "... our U-Net++, but have trained multiple ...". Line 167, "... the best possible output. After this training, we ...". Line 176, "... section presents the influence of increasing the ...". Line 254 & 255, "... receive an automatic diagnosis almost ...". Line 268, "... inner and outer layers that humans ...". Line 271 & 272, "For this reason, many small heart images to learn the segment properly have not been available." Lines 279 & 280, "... neural network that would generally be better to other ...". Lines 283 & 284, "... difference in the results from different hospitals." Line 301, "... approach will mostly benefit the patient." Line 304, "... low trabeculae. In future studies, we plan to ...".
Comments on the Quality of English LanguageThe manuscript is well written; however, verb tense should be evaluated in some sections of the manuscript.
Reviewer 4 Report
Comments and Suggestions for Authors
The aim of the work presented was to increase the reliability of LVNC (Left Ventricular Noncompaction Cardiomyopathy) detection by improving the segmentation of the left ventricle in cardiac MR images. The authors used data sets from three different hospitals with very different patient numbers and different qualities of LVNC diagnoses. Three main methods were compared, first using full MR images rather than with reduced resolution, second a clustering algorithm to eliminate neural network hallucinations and further more an advanced network architecture (AttU-Net, MAS-UNet and U-Net++). Among other things, a slight improvement in the average Dice score was achieved, but although differences between the hospital data sets were noted. The authors interpret the results in the way to provide with it a robust system for determining LVNC that can allow cardiologists with a diagnosis in a short time. The authors expect the performance of the system to be further improved by supplementing the currently unbalanced image data sets with additional data sets in the future.
The paper is well structured, written in a comprehensible manner and the limitations of the statements are also shown.
Congratulations on this clear and well-structured work! Additionally, I wish you much success in expanding your data sets with new images to address your received imbalance to images of left ventricles with extensive trabeculae.
Round 2
Reviewer 1 Report
Comments and Suggestions for Authors
No further question.
Comments on the Quality of English LanguageMinor refinement